# Can a meditation app help my sleep? A cross-sectional survey of Calm users

**Jennifer Huberty**[1]☯*, **Megan E. Puzia**[2]☯, **Linda Larkey**[3]☯, **Ana-Maria Vranceanu**[4]☯,
**Michael R. Irwin**[5]☯

**1** College of Health Solutions, Arizona State University, Phoenix, Arizona, United States of America,
**2** Behavioral Research and Analytics, LLC, Salt Lake City, Utah, United States of America, **3** Edson College
of Nursing and Health Innovation, Center for Health Promotion and Disease Prevention, Arizona State
University, Phoenix, Arizona, United States of America, **4** Department of Psychiatry, Integrated Brain Health
Clinical and Research Program, Massachusetts General Hospital/Harvard Medical School, Boston,
Massachusetts, United States of America, **5** Department of Psychiatry and Biobehavioral Sciences, Cousins
Center for Psychoneuroimmunology, Mindful Awareness Research Center, Jane and Terry Semel Institute for
Neuroscience and Human Behavior, David Geffen School of Medicine, University of California, Los Angeles,
California, United States of America

☯ These authors contributed equally to this work.
* JHuberty@asu.edu

SPAIN

**Data Availability Statement:** Data used in this
study are available from the OSF repository at
https://doi.org/10.17605/OSF.IO/Z5AJS (DOI: 10.
17605/OSF.IO/Z5AJS).

## Abstract

Use of mindfulness mobile apps has become popular, however, there is little information
about subscribers' perceptions of app content and its impact on sleep and mental health.
The purpose of this study was to survey subscribers to Calm, a popular mindfulness medita-
tion app, to explore perceived improvements in sleep and mental health, evaluate what com-
ponents of the app were associated with improvements in sleep and mental health, and
determine whether improvements differed based on sleep quality. Calm subscribers who
had used a sleep-related component in the last 90 days completed a Web-based investiga-
tor-developed survey and the Pittsburgh Sleep Quality Index. The survey included questions
about using Calm for sleep, sleep disturbances, mental health diagnoses (i.e., anxiety,
depression, PTSD) and perceived impacts of the app. Participants reported on the extent to
which they felt that using Calm had improved their sleep and mental health. Most partici-
pants reported sleep disturbance, and almost half reported a mental health diagnosis. The
majority of participants reported that using Calm helped them fall asleep, stay asleep, and
get restful sleep. All sleep components were associated with perceived improvements in
sleep disturbance. Severity of sleep disturbance moderated relationships between using
Calm components and reporting improved sleep. Among subscribers with mental health
diagnoses, most reported that Calm helped improve symptoms. Perceived improvement in
anxiety and depression was associated with using Calm's meditation components but not
Sleep Stories or music/soundscapes. Severity of sleep disturbance did not moderate rela-
tionships between using Calm components and reporting mental health improvements.
Given the accessibility of app-based meditation, research is needed to evaluate the efficacy
of meditation apps to improve sleep disturbance. While some sleep content may be helpful
for sleep, more research is needed to test what specific content affects mental health.

**Funding:** The author(s) received no specific funding for this work.

**Competing interests:** JH is currently the Director of Science at Calm. JH has been conducting research with Calm as a partner almost 5 years before becoming the Director of Science and the Scientific Advisory Board. AMV, LL, and MI are members of Calm's Scientific Advisory Board and are independent from Calm leadership. Their role is to ensure the quality of Calm's science. There are no financial incentives from the growth of Calm to any author. This does not alter our adherence to PLOS ONE policies on sharing data and materials.

# Introduction

## Background/Rationale

Adequate sleep is essential for good mental and physical health [1], yet more than 60% of Americans report sleep disturbances including trouble falling or staying asleep, sleeping excessively, disturbed sleep-wake schedules, and restless sleep [2–4]. In 10% of these individuals sleep disturbances are severe or chronic meeting the diagnostic criteria for insomnia [5]. Sleep disturbance is both a risk factor for and a comorbidity of chronic diseases [6] and emotional distress (i.e., depression, anxiety, post-traumatic stress disorder; PTSD); impacting well-being and quality of life [6, 7].

Over the last decade, pharmacotherapy and behavioral therapies (i.e., cognitive behavioral therapy, relaxation techniques) have emerged as the most widely used treatments for sleep disturbance [4]. Although effective, both approaches have limitations. Even though sleep medications can help in the short term, they carry risk of dependence if used long term, their effectiveness tends to decline with extended use, and their benefits diminish entirely after drug discontinuation [8, 9]. Similarly, although behavioral therapies have been shown to be effective (e.g., CBT-I) [10], they are time- and cost-intensive, and require highly trained therapists. Further, such treatments are not always covered by insurance, making it inaccessible to many individuals in need of treatment [11]. Highly accessible, convenient, safe and effective treatments are needed in order to effectively address this growing public health problem.

One potential solution is mindfulness meditation. Mindfulness meditation is the practice of attending to moment-by-moment experiences, thoughts, and emotions without judgment [12, 13]. Prior research, including systematic reviews, shows that mindfulness meditation programs are efficacious for reducing sleep disturbance [4, 8, 14, 15]. Mindfulness meditation may improve sleep through several mechanisms including increased awareness, acceptance, decreased ruminative thoughts and emotional reactivity, and promotion of an impartial reappraisal of salient experiences, which are all associated with sleep disturbance [16, 17].

Mindfulness meditation can be effectively delivered using a mobile app [18–20]. A recent survey of subscribers to the meditation mobile app, Calm showed that sleep disturbances were one of the most common reasons for downloading the app, and that individuals with sleep disturbances used the app more often and were more likely to use sleep content [21]. In response to the epidemic of sleep complaints and insomnia, Calm and other mobile app companies (e.g., Insight Timer, Headspace) have more recently diversified their content to include meditations and soundscapes specifically for sleep. Calm has created Sleep Stories, narrated fictional tales that use mindfulness-inspired techniques (e.g., breath-focused meditation, sensory meditations) that promote experiential awareness (e.g., of thoughts, positive emotions, sights, sounds) to disrupt the cycle of rumination, and lower pre-sleep arousal to help individuals fall and stay asleep [4]. Headspace has designed Sleepcasts, which are audio-guided exercises similar to Sleep Stories but with less narration. Both Sleep Stories and Sleepcasts emphasize the sensory experience through detailed descriptions and ambient sound effects.

Although mindfulness apps have become popular sleep aids within the general population, there remains a knowledge gap related to content subscribers use for sleep (i.e., general meditation vs. sleep content) and how this impacts their sleep (i.e., falling asleep, staying asleep, not waking too early, and getting restful sleep). We also do not know if individuals who use mobile meditation apps for sleep also experience mental and physical health benefits known to be associated with sleep (e.g., depression, anxiety disorders, and PTSD) [22]. Additionally, if use of meditation apps is helpful for improving sleep and mental health outcomes, we do not know whether all users benefit equally. For example, it is unclear whether the associations

between using Calm and reporting improvements in sleep will be similar for individuals with more severe disturbance compared to those with only mild sleep disturbance. Thus, there is a need to explore how consumers engage with mobile meditation apps for sleep, what aspects of the app are beneficial to sleep disturbed users, and whether these perceived benefits depend on the severity of self-reported sleep disturbance.

## Objectives

The purpose of this study was to conduct a cross-sectional survey of subscribers to Calm, one of the more popular consumer-based meditation mobile apps. First, we examined the extent to which those who use the app for sleep report improvements in sleep and mental health. Second, we assessed whether improvements in sleep and mental health were differentially related to use of specific components of the app. Finally, we explored whether these improvements based on the specific components used depended on (i.e., were moderated by) sleep quality.

## Methods and materials

The present analyses are secondary analyses part of a larger cross-sectional survey [23] that explored Calm subscribers' use of the app for sleep, reported in accordance with the STROBE cross-sectional reporting guidelines [24]. There was no patient or public involvement in the production of this study. The larger cross-sectional survey was approved by an Institutional Review Board at Arizona State University (STUDY00011083). All participants signed an approved electronic informed consent document prior to participation.

### Participants/recruitment

Participants were Calm subscribers who 1) completed at least one session of Calm using a sleep-related component (sleep meditations, Sleep Stories, soundscapes) in the last 90 days, and 2) opened at least one email from Calm within the last 90 days. Subscribers meeting these criteria were sent an email asking if they would like to participate in an online survey with questions and feedback related to their experience with Calm to help improve the Calm app. Those who completed the survey were entered into a drawing for one of two $99 Amazon gift cards. Potential participants used a link within the recruitment email to verify that they were 1) at least 18 years old and 2) were able to read and understand questions in English. Those eligible were then directed to an electronic consent and the survey questions.

### Measures

Participants completed a demographics survey, the Pittsburgh Sleep Quality Index (PSQI) and an investigator-developed quantitative survey. Severity of self-reported sleep disturbance was assessed using the Pittsburgh Sleep Quality Index (PSQI) a 19-item questionnaire that measures sleep habits, quality, and disturbances over the past month [25]. This widely used measure has good psychometric properties across a range of clinical and nonclinical populations; including good internal ($\alpha = .83$) and test-retest reliability ($r = .85$) as well as concurrent and discriminative validity when compared to clinical evaluation and other self-report sleep measures [26, 27]. Dimensions of sleep disturbance were calculated based on a validated 3-factor model [28].

To evaluate use patterns and perceptions regarding the Calm app, an investigator-developed survey was administered. Consistent with the assessment of sleep quality using the PSQI, questions were asked if participants were experiencing sleep disturbances at the time they download Calm (i.e., difficulty with falling asleep, staying asleep, waking too early, or getting

restful sleep. App usage patterns were evaluated with questions about how often participants used Calm in general (i.e., number of times per week) and how often they used specific Calm components at night for sleep (i.e., Sleep Stories, sleep meditations, non-sleep meditations, and music/soundscapes). Perceived benefit of using Calm was explored by asking participants about the extent to which they believed Calm usage had helped to improve their sleep disturbance. Specifically, participants were asked whether using Calm *very much improved* (= 3), *somewhat improved* (= 2), or caused *no noticeable improvement* (= 1) in their sleep. The questions for mental health conditions (i.e., anxiety, depression, PTSD) followed the same format (i.e., diagnosis and then perceptions of improvement).

## Statistical analysis

Analyses were conducted using IBM SPSS 26.0. Frequencies and descriptive statistics were used to characterize the sample and to examine app usage patterns. Correlations were used to examine within-participant associations between sleep quality and app usage. Associations between app usage and improvements in sleep and mental health were assessed in stepwise models using the PLUM ordinal regression procedure. At step 1, models regressed self-reported improvements in sleep and mental health outcomes onto app usage frequency. To examine whether severity of sleep disturbance moderated the relationship between usage and perceived improvements, PSQI score usage × PSQI were added as predictors in Step 2. Goodness of fit was evaluated via -2 log-likelihoods, Pearson's chi-square, and pseudo $R^2$ statistics. To correct for multiple testing, Bonferroni-adjusted p-values were used to determine significance. Given the likely correlations between use of different Calm components, individual-component models were followed up with multivariate models that included all components as simultaneous predictors. To reduce risk of over-fitting, only significant PSQI score usage × PSQI interactions were retained in multivariate models.

# Results

There were 366,173 subscribers who received an invitation to participate. Of those, 5.6% ($n$ = 19,341) accessed the survey, 4.0% ($n$ = 14,642) signed the informed consent, and 3.0% ($n$ = 11,095) viewed all of the survey questions (i.e., completion rate = 75.8% of consented participants). Participants were included in the present analyses if they had 1) used at least one Calm component for sleep and 2) responded to questions about experiencing sleep disturbance and/or mental health diagnoses at the time of app download. The final sample included 9,907 participants.

## Demographic characteristics

Sample demographics are presented in Table 1. On average, participants were 47 years old ($SD$ = 15.4). The majority identified as female (85.3%, $n$ = 8,230), White (83.9%), and non-Hispanic (94.7%). Most participants had received a higher-education degree (84.5%) and had full- or part-time employment (74.7%)The median household income was $80,000 per year.

## Sleep and health characteristics

Most participants (89.7%, $n$ = 8,886) reported having some type of sleep disturbance when they initially downloaded Calm, and most reported more than one type of disturbance. The most common areas of sleep disturbance were falling asleep (65.5%, $n$ = 6,489), staying asleep (48.5%, $n$ = 4,807), and getting restful sleep (42.0%, $n$ = 4,159), and many individuals experienced more than one sleep disturbance (50.6%, $n$ = 5,015). Only 15.2% ($n$ = 1,503) reported

**Table 1. Demographic characteristics of the sample (N = 9,907).**

| Category | | *n* (%) |
|---|---|---|
| Race (*N* = 9,403) | | |
| | White, European American, or Caucasian | 8315 (83.9) |
| | Asian or Asian-American | 315 (3.2) |
| | Black, African American, or Native African | 248 (2.5) |
| | American Indian or Alaska Native | 85 (0.9) |
| | Native Hawaiian or Pacific Islander | 16 (0.2) |
| | Bi-racial or Multi-racial | 282 (2.8) |
| | Other | 424 (4.3) |
| Ethnicity (*N* = 9,604) | | |
| | Non-Hispanic | 9097 (94.7) |
| | Hispanic | 507 (5.3) |
| Gender (*N* = 9,754) | | |
| | Female | 8320 (85.3) |
| | Male | 1400 (14.4) |
| | Other | 34 (0.3) |
| Employment (*N* = 9,681) | | |
| | Full-time employment | 6027 (62.3) |
| | Part-time employment | 1201 (12.4) |
| | Unemployed | 342 (3.5) |
| | Disability | 243 (2.5) |
| | Full-time student | 281 (2.9) |
| | Other | 1587 (16.4) |
| Education (*N* = 9,707) | | |
| | 11th grade or less | 40 (0.4) |
| | High school or GED | 468 (4.7) |
| | Some College | 1001 (10.3) |
| | Two-year college/technical school | 957 (9.9) |
| | Bachelors degree or equivalent | 3655 (37.7) |
| | Graduate degree or equivalent | 3450 (35.5) |
| | Other | 136 (1.4) |

that they experienced difficulties related to waking too early. Reports about recent sleep quality (i.e., based on the PSQI) also indicated high rates of sleep disturbance (see Table 2). The mean PSQI total score for the sample was 7.0 (*SD* = 3.6), and 61.5% of participants (*n* = 6,093) scored above the established PSQI cutoff (i.e., total score > 5; Buysse et al., 1989), classifying them as "poor sleepers."

**Table 2. PSQI total and component scores.**

| PSQI Factor | N | M (SD) |
|---|---|---|
| Sleep Efficiency | 9,248 | 1.5 (1.8) |
| Perceived Sleep Quality | 9,893 | 3.2 (2.0) |
| Daily Disturbances | 9,556 | 2.4 (1.0) |
| Total PSQI Score | 9,907 | 7.0 (3.6) |

*Note*. Possible ranges for factor scores are 0 to 6 for Sleep Efficiency and Daily Disturbances, and 0 to 9 for Perceived Sleep Quality. For all factors, higher scores indicate more severe sleep disturbance.

**Table 3. Correlations between frequencies of using individual Calm components.**

| Component | Sleep Stories r, p | Sleep meditations r, p | General meditations r, p | Music r, p | PSQI r, p | N | Times used/week M (SD) |
|---|---|---|---|---|---|---|---|
| Overall | .53, < .001 | .27, < .001 | .36, < .001 | .29, < .001 | -.04, < .001 | 9,894 | 5.1 (2.0) |
| Sleep Stories | | .16, < .001 | .01, .32 | .14, < .001 | .04, < .001 | 9,050 | 3.8 (2.4) |
| Sleep meditations | | | .47, < .001 | .33, < .001 | .08, < .001 | 6,501 | 2.0 (2.1) |
| General meditations | | | | .26, < .001 | -.06, < .001 | 6,091 | 2.7 (2.5) |
| Music | | | | | .03, .01 | 6,558 | 2.4 (2.4) |

Almost half of participants reported having been diagnosed with at least one mental health condition when they initially downloaded Calm (44.3%, $n = 4,386$), the most common being anxiety disorders (29.6%, $n = 2,930$) and depression (26.8%, $n = 2,657$). Additionally, 8.0% ($n = 790$) reported a diagnosis of PTSD.

## App and component engagement

On average, participants used Calm five times per week (see Table 3). Sleep Stories were the most commonly used component at night for sleep. There were significant moderate correlations between reported usage frequencies across almost all individual components, except that frequency of using Sleep Stories was not related to the frequency of using general (i.e., not sleep-specific) meditations at night. PSQI scores were also correlated with app usage, but these effects were small.

## Improvements in sleep

The majority of participants reported that using Calm helped them fall asleep (90.0% reported *somewhat* or *very much* improved), stay asleep (69.7%), and get restful sleep (79.2%), but the small proportion of individuals who reported problems waking up too early did not notice improvement in this area (39.6%; see Table 4).

At Step 1, frequency of using Calm at night was positively associated with perceived improvements in all aspects of sleep (i.e., falling asleep, staying asleep, getting restful sleep, and not waking too early; see Table 5). At Step 2, there were no significant main effects of PSQI, but there were significant interactions in which PSQI scores moderated the relationship between the frequency of using Calm and improvements in staying asleep and getting restful sleep, such that participants who reported more severe sleep disturbance benefitted more from using Calm with regard to improvements in staying asleep and getting restful sleep (see Table 6). Although using Calm was also associated with improvements in falling asleep and waking too early, these improvements did not differ based on severity of sleep disturbance.

## Components of the app and perceived sleep improvements

The frequency of using Sleep Stories, sleep meditations, and music/soundscapes was associated with improvement in falling asleep, but using general meditations at night was not (see

**Table 4. Frequencies of reported improvements in sleep disturbance.**

| Type of sleep disturbance | N | No noticeable improvement, n (%) | Somewhat improved, n (%) | Very much improved, n (%) |
|---|---|---|---|---|
| Fall asleep | 9,671 | 772 (8.0) | 3,151 (32.6) | 5,748 (59.4) |
| Stay asleep | 9,250 | 2,807 (30.3) | 4,109 (44.4) | 2,334 (25.2) |
| Getting restful sleep | 8,514 | 1,899 (20.8) | 4,619 (50.5) | 2,624 (28.7) |
| Not waking up too early | 9,142 | 5,144 (60.4) | 2,440 (28.7) | 930 (10.9) |

**Table 5. Parameter estimates for regressions models of improvements in sleep disturbance based on frequency of using Calm and its components (i.e., Step 1).**

| | | Falling asleep | | | Staying asleep | | | Restful sleep | | | Waking early | | |
|---|---|---|---|---|---|---|---|---|---|---|---|---|---|
| Model | N | Est. (SE) | p | N | Est. (SE) | p | N | Est. (SE) | p | N | Est. (SE) | p |
| Calm, overall | 9,658 | | | 9,237 | | | 9,129 | | | 8,502 | | |
| Intercept [threshold = 0] | | -1.39 (0.05) | < .001 | | 0.24 (0.05) | < .001 | | -0.29 (0.05) | < .001 | | 1.40 (0.06) | < .001 |
| Intercept [threshold = 1] | | 0.73 (0.05) | < .001 | | 2.22 (0.05) | < .001 | | 2.00 (0.05) | < .001 | | 3.09 (0.06) | < .001 |
| Usage frequency | | 0.22 (0.01) | < .001 | | 0.21 (0.01) | < .001 | | 0.20 (0.01) | < .001 | | 0.18 (0.01) | < .001 |
| Sleep Stories | 8,868 | | | 8,489 | | | 8,388 | | | 7,806 | | |
| Intercept [threshold = 0] | | -1.69 (0.04) | < .001 | | -0.21 (0.03) | < .001 | | -0.79 (0.04) | < .001 | | 0.87 (0.04) | < .001 |
| Intercept [threshold = 1] | | 0.71 (0.04) | < .001 | | 1.81 (0.04) | < .001 | | 1.53 (0.04) | < .001 | | 2.56 (0.05) | < .001 |
| Usage frequency | | 0.32 (0.01) | < .001 | | 0.18 (0.01) | < .001 | | 0.16 (0.01) | < .001 | | 0.11 (0.01) | < .001 |
| Sleep meditations | 6,382 | | | 6,178 | | | 6,153 | | | 5,769 | | |
| Intercept [threshold = 0] | | -2.32 (0.05) | < .001 | | -0.58 (0.03) | < .001 | | -1.16 (0.03) | < .001 | | 0.70 (0.03) | < .001 |
| Intercept [threshold = 1] | | -0.15 (0.03) | < .001 | | 1.46 (0.03) | < .001 | | 1.20 (0.03) | < .001 | | 2.44 (0.05) | < .001 |
| Usage frequency | | 0.13 (0.01) | < .001 | | 0.19 (0.01) | < .001 | | 0.16 (0.01) | < .001 | | 0.18 (0.01) | < .001 |
| General meditations | 5,983 | | | 5,787 | | | 5,760 | | | 5,404 | | |
| Intercept [threshold = 0] | | -2.46 (0.05) | < .001 | | -0.71 (0.03) | < .001 | | -1.22 (0.04) | < .001 | | 0.64 (0.04) | < .001 |
| Intercept [threshold = 1] | | -0.35 (0.03) | < .001 | | 1.27 (0.04) | < .001 | | 1.10 (0.04) | < .001 | | 2.36 (0.05) | < .001 |
| Usage frequency | | 0.01 (0.01) | .294 | | 0.09 (0.01) | < .001 | | 0.09 (0.01) | < .001 | | 0.13 (0.01) | < .001 |
| Music/soundscapes | 6,446 | | | 6,224 | | | 6,186 | | | 5,776 | | |
| Intercept [threshold = 0] | | -2.38 (0.05) | < .001 | | -0.63 (0.03) | < .001 | | -1.21 (0.03) | < .001 | | 0.65 (0.03) | < .001 |
| Intercept [threshold = 1] | | -0.21 (0.03) | < .001 | | 1.38 (0.03) | < .001 | | 1.09 (0.03) | < .001 | | 2.34 (0.04) | < .001 |
| Usage frequency | | 0.09 (0.01) | < .001 | | 0.13 (0.01) | < .001 | | 0.10 (0.01) | < .001 | | 0.12 (0.01) | < .001 |

Table 5) (Step 1). However, all sleep components were associated with improvements in staying asleep, getting restful sleep, and not waking up too early. Perceived improvement in falling asleep was positively associated with the use of Sleep Stories, sleep meditations and music/soundscapes, but general meditations; however, all components were associated with improvements in staying asleep, getting restful sleep, and not waking too early (see Table 5).

At Step 2, the conditional main effects of frequency of usage and PSQI scores were maintained after accounting for possible interactions between frequency of use and severity of sleep disturbance; however, there were also several significant interactions between usage and PSQI scores (see Table 6). Specifically, PSQI scores moderated the relationships between using Sleep Stories and sleep meditations and reporting improvements in falling asleep, such that these components appeared to be more beneficial for participants with more severe sleep disturbance. Significant moderation effects also suggested that those with more severe sleep disturbance benefitted more from using Sleep Stories, sleep meditations, and general meditations to help them to stay asleep. The relationships between using Calm components and perceived improvements in getting restful sleep was were also moderated by PSQI, such that all Calm sleep components appeared to be more beneficial for participants with more severe sleep difficulties. Results also showed significant moderation effects suggesting that using meditations (both sleep and general) was also more beneficial for helping those with more severe sleep disturbance not wake up too early.

When all components were analyzed together (i.e., when controlling for using other Calm components for sleep), results showed all individual components of Calm still uniquely contributed to perceived improvements in all aspects of sleep (see Table 7). Several significant moderation effects were retained in the combined model. Results indicated that those with more severe sleep disturbance benefitted more from using Sleep Stories to help them fall

**Table 6. Parameter estimates for regressions models of improvements in sleep disturbance based on frequency of using Calm and its components, PSQI scores, and the interaction between frequency of use and PSQI (i.e., Step 2).**

| | | Falling asleep | | | Staying asleep | | | Restful sleep | | | Waking early | | |
|---|---|---|---|---|---|---|---|---|---|---|---|---|---|
| Model | | N | Est. (SE) | p | N | Est. (SE) | p | N | Est. (SE) | p | N | Est. (SE) | p |
| Calm, overall | | 9,658 | | | 9,237 | | | 9,129 | | | 8,502 | | |
| | Intercept [threshold = 0] | | -1.64 (0.12) | < .001 | | 0.18 (0.12) | .130 | | -0.34 (0.12) | .004 | | 1.22 (0.14) | < .001 |
| | Intercept [threshold = 1] | | 0.50 (0.12) | < .001 | | 2.20 (0.12) | < .001 | | 2.05 (0.12) | < .001 | | 2.94 (0.15) | < .001 |
| | Usage frequency | | 0.24 (0.02) | < .001 | | 0.29 (0.02) | < .001 | | 0.34 (0.02) | < .001 | | 0.22 (0.03) | < .001 |
| | PSQI | | -0.02 (0.01) | .102 | | -0.00 (0.01) | .842 | | 0.01 (0.01) | .594 | | -0.02 (0.02) | .360 |
| | Usage x PSQI | | -0.01 (0.00) | .104 | | -0.01 (0.00) | < .001 | | -0.02 (0.00) | < .001 | | -0.01 (0.00) | .069 |
| Sleep Stories | | 8,868 | | | 8,489 | | | 8,388 | | | 7,806 | | |
| | Intercept [threshold = 0] | | -1.90 (0.09) | < .001 | | -0.51 (0.08) | < .001 | | -1.38 (0.09) | < .001 | | 0.49 (0.09) | < .001 |
| | Intercept [threshold = 1] | | 0.53 (0.09) | < .001 | | 1.55 (0.09) | < .001 | | 1.05 (0.09) | < .001 | | 2.20 (0.10) | < .001 |
| | Usage frequency | | 0.45 (0.02) | < .001 | | 0.26 (0.02) | < .001 | | 0.23 (0.02) | < .001 | | 0.13 (0.02) | < .001 |
| | PSQI | | -0.02 (0.01) | .034 | | -0.03 (0.01) | .003 | | -0.07 (0.01) | < .001 | | -0.04 (0.01) | < .001 |
| | Usage x PSQI | | -0.02 (0.00) | < .001 | | -0.01 (0.00) | < .001 | | -0.01 (0.00) | < .001 | | -0.00 (0.00) | .159 |
| Sleep meditations | | 6,382 | | | 6,178 | | | 6,153 | | | 5,769 | | |
| | Intercept [threshold = 0] | | -2.72 (0.09) | < .001 | | -1.10 (0.08) | < .001 | | -1.89 (0.08) | < .001 | | 0.32 (0.08) | < .001 |
| | Intercept [threshold = 1] | | -0.50 (0.08) | < .001 | | 1.00 (0.08) | < .001 | | 0.60 (0.08) | < .001 | | 2.09 (0.09) | < .001 |
| | Usage frequency | | 0.24 (0.03) | < .001 | | 0.28 (0.03) | < .001 | | 0.26 (0.03) | < .001 | | 0.25 (0.03) | < .001 |
| | PSQI | | -0.05 (0.01) | < .001 | | -0.06 (0.01) | < .001 | | -0.09 (0.01) | < .001 | | -0.05 (0.01) | < .001 |
| | Usage x PSQI | | -0.01 (0.00) | < .001 | | -0.01 (0.00) | < .001 | | -0.01 (0.00) | < .001 | | -0.01 (0.00) | .010 |
| General meditations | | 5,983 | | | 5,787 | | | 5,760 | | | 5,404 | | |
| | Intercept [threshold = 0] | | -3.02 (0.10) | < .001 | | -1.23 (0.08) | < .001 | | -1.96 (0.09) | < .001 | | 0.31 (0.09) | .001 |
| | Intercept [threshold = 1] | | -0.85 (0.09) | < .001 | | 0.78 (0.08) | < .001 | | 0.48 (0.08) | < .001 | | 2.05 (0.10) | < .001 |
| | Usage frequency | | -0.01 (0.02) | .684 | | 0.12 (0.02) | < .001 | | 0.14 (0.02) | < .001 | | 0.16 (0.02) | < .001 |
| | PSQI | | -0.06 (0.01) | < .001 | | -0.06 (0.01) | < .001 | | -0.08 (0.01) | < .001 | | -0.04 (0.01) | .001 |
| | Usage x PSQI | | 0.00 (0.00) | .750 | | -0.01 (0.00) | .045 | | -0.01 (0.00) | .003 | | -0.01 (0.00) | .043 |
| Music/soundscapes | | 6,446 | | | 6,224 | | | 6,186 | | | 5,776 | | |
| | Intercept [threshold = 0] | | -2.85 (0.09) | < .001 | | -1.25 (0.08) | < .001 | | -1.98 (0.08) | < .001 | | 0.15 (0.08) | .080 |
| | Intercept [threshold = 1] | | -0.62 (0.08) | < .001 | | 0.80 (0.08) | < .001 | | 0.44 (0.08) | < .001 | | 1.86 (0.09) | < .001 |
| | Usage frequency | | 0.11 (0.02) | < .001 | | 0.14 (0.02) | < .001 | | 0.14 (0.02) | < .001 | | 0.11 (0.02) | < .001 |
| | PSQI | | -0.05 (0.01) | < .001 | | -0.08 (0.01) | < .001 | | -0.09 (0.01) | < .001 | | -0.06 (0.01) | < .001 |
| | Usage x PSQI | | -0.00 (0.00) | .276 | | -0.00 (0.00) | .544 | | -0.01 (0.00) | .039 | | 0.00 (0.00) | .678 |

asleep, stay asleep, and get more restful. Those with more severe sleep disturbance also benefitted more from using sleep meditations with regard to falling asleep, whereas general meditations were more beneficial for helping those with more severe sleep disturbance stay asleep, get more restful sleep, and not wake up too early.

## Improvements in mental health

Among those who reported having a mental health diagnosis when they initially downloaded Calm, most reported that using Calm had helped to improve their condition or their ability to manage their symptoms (see Table 8). The highest rate of improvement was in anxiety disorders (90.5% reported *somewhat* or *very much* improved), followed by depression (80.3%), and then PTSD (77.2%).

Using Calm more frequently was associated with greater perceived improvements in anxiety, depression, and PTSD (Step 1; see Table 9), even after and accounting for PSQI scores as a potential moderator (Step 2; see Table 10). At Step 2 there were significant conditional main

**Table 7. Parameter estimates for regressions models of improvements in sleep difficulties based on PSQI scores and frequency of using all Calm components.**

| | Model | | | | | | | |
|---|---|---|---|---|---|---|---|---|
| | Falling asleep (N = 4,456) | | Staying asleep (N = 4,334) | | Restful sleep (N = 4,321) | | Waking too early (N = 4,091) | |
| Predictor | Est. (SE) | p | Est. (SE) | p | Est. (SE) | p | Est. (SE) | p |
| Intercept [threshold = 0] | -1.60 (0.14) | < .001 | -0.16 (0.14) | .233 | -1.07 (0.14) | < .001 | 0.91 (0.12) | < .001 |
| Intercept [threshold = 1] | 0.79 (0.14) | < .001 | 2.02 (0.14) | < .001 | 1.47 (0.14) | < .001 | 2.72 (0.13) | < .001 |
| Sleep Stories usage | 0.46 (0.03) | < .001 | 0.26 (0.03) | < .001 | 0.21 (0.03) | < .001 | 0.10 (0.01) | < .001 |
| Sleep meditation usage | 0.21 (0.04) | < .001 | 0.16 (0.03) | < .001 | 0.15 (0.04) | < .001 | 0.15 (0.04) | < .001 |
| General meditation usage | -0.04 (0.02) | .004 | 0.09 (0.03) | .003 | 0.09 (0.03) | .002 | 0.15 (0.03) | < .001 |
| Music/soundscape usage | 0.05 (0.02) | .002 | 0.08 (0.03) | < .001 | 0.06 (0.03) | .042 | 0.07 (0.01) | < .001 |
| PSQI | -0.01 (0.02) | .495 | -0.02 (0.02) | .310 | -0.05 (0.02) | .007 | -0.32 (0.01) | .018 |
| Sleep Stories x PSQI | -0.02 (0.00) | < .001 | -0.01 (0.00) | < .001 | -0.01 (0.00) | .006 | – | – |
| Sleep meditation x PSQI | -0.01 (0.00) | .006 | -0.00 (0.00) | .409 | -0.01 (0.00) | .208 | -0.01 (0.00) | .179 |
| General meditation x PSQI | – | – | -0.01 (0.00) | .021 | -0.01 (0.00) | .049 | -0.01 (0.00) | .042 |
| Music/soundscapes x PSQI | – | – | – | – | -0.00 (0.00) | .883 | – | – |

effects of PSQI in which PQSI was negatively associated with improvements in mental health diagnoses, but PSQI did not significantly moderate the associations between frequency of use and mental health improvements.

## Components of the app and mental health improvements

Prior to controlling for severity of sleep disturbance (Step 1), the frequency of using sleep meditations, general meditations, and music/soundscapes was associated with perceived improvements in anxiety, depression and PTSD (see Table 9). After accounting for potential moderation effects at Step 2, meditations (sleep and general) were still associated with improvements in depression and anxiety, but no longer associated with PTSD (see Table 10). Using music/soundscapes was still associated with improvements in all mental health diagnoses. There were no significant moderation effects. The frequency of using Sleep Stories was not related to improvements in any mental health diagnosis in any model.

As observed in analyses of individual Calm components, in combined analyses when all components were assessed together, using sleep meditations was still associated with improvements in anxiety, but not depression (see Table 11). Conversely, the frequency of using general meditations was positively associated with all mental health diagnoses. After controlling for the use of other components, using music/soundscapes was no longer associated with improvements in depression, anxiety, or PTSD. Using Sleep Stories was not associated with reported improvements in mental health.

## Discussion

The purpose of this study was to conduct a cross-sectional survey in subscribers of the mobile app Calm to examine the extent to which those who use the app for sleep report improvements

**Table 8. Frequencies of reported improvements in mental health diagnoses.**

| Diagnosis | N | No noticeable improvement, n (%) | Somewhat improved, n (%) | Very much improved, n (%) |
|---|---|---|---|---|
| Anxiety | 2,868 | 273 (9.5) | 1,541 (53.7) | 1,054 (36.8) |
| Depression | 2,587 | 511 (19.8) | 1,382 (53.4) | 694 (26.8) |
| PTSD | 771 | 176 (22.8) | 401 (52.0) | 194 (25.2) |

**Table 9. Parameter estimates for regressions models of improvements in mental health diagnoses based on frequency of using Calm and its components (i.e., Step 1).**

| | Predictor | Anxiety | | | Depression | | | PTSD | | |
|---|---|---|---|---|---|---|---|---|---|---|
| | | N | Est. (SE) | p | N | Est. (SE) | p | N | Est. (SE) | p |
| Calm, overall | | 2,865 | | | 2,584 | | | 771 | | |
| | Intercept [threshold = 0] | | -1.25 (0.11) | < .001 | | -0.40 (0.11) | < .001 | | -0.03 (0.19) | .877 |
| | Intercept [threshold = 1] | | 1.65 (0.11) | < .001 | | 2.10 (0.12) | < .001 | | 2.40 (0.21) | < .001 |
| | Usage frequency | | 0.21 (0.02) | < .001 | | 0.20 (0.02) | < .001 | | 0.24 (0.04) | < .001 |
| Sleep Stories | | 2,615 | | | 2,383 | | | 706 | | |
| | Intercept [threshold = 0] | | -2.25 (0.09) | < .001 | | -1.33 (0.08) | < .001 | | -1.02 (0.14) | < .001 |
| | Intercept [threshold = 1] | | 0.54 (0.07) | < .001 | | 1.05 (0.08) | < .001 | | 1.38 (0.15) | < .001 |
| | Usage frequency | | 0.00 (0.02) | .999 | | 0.01 (0.02) | .473 | | 0.06 (0.03) | .054 |
| Sleep meditations | | 1,997 | | | 1,782 | | | 565 | | |
| | Intercept [threshold = 0] | | -2.31 (0.09) | < .001 | | -1.36 (0.07) | < .001 | | -1.21 (0.13) | < .001 |
| | Intercept [threshold = 1] | | 0.62 (0.06) | < .001 | | 1.13 (0.07) | < .001 | | 1.21 (0.13) | < .001 |
| | Usage frequency | | 0.11 (0.02) | < .001 | | 0.12 (0.02) | < .001 | | 0.08 (0.03) | .019 |
| General meditations | | 1,872 | | | 1,660 | | | 524 | | |
| | Intercept [threshold = 0] | | -2.19 (0.10) | < .001 | | -1.28 (0.08) | < .001 | | -1.05 (0.14) | < .001 |
| | Intercept [threshold = 1] | | 0.81 (0.07) | < .001 | | 1.37 (0.08) | < .001 | | 1.40 (0.15) | < .001 |
| | Usage frequency | | 0.17 (0.02) | < .001 | | 0.19 (0.02) | < .001 | | 0.13 (0.03) | < .001 |
| Music/soundscapes | | 1,951 | | | 1,793 | | | 555 | | |
| | Intercept [threshold = 0] | | -2.14 (0.09) | < .001 | | -1.33 (0.07) | < .001 | | -0.27 (0.12) | .026 |
| | Intercept [threshold = 1] | | 0.63 (0.07) | < .001 | | 1.10 (0.07) | < .001 | | 2.40 (0.15) | < .001 |
| | Usage frequency | | 0.08 (0.02) | < .001 | | 0.06 (0.02) | .001 | | 0.02 (0.02) | .377 |

in sleep and mental health. We also assessed whether improvements in sleep and mental health were differentially related to use of specific components of the app, and explored whether these improvements based on using specific components used depended on sleep quality. explore the extent to which those who use the app for sleep report improvements in sleep and mental health. We also explored which components of the app are associated with sleep and mental health improvements, and whether the improvements associated with using the app were different based on self-reported sleep quality (as measured by the PSQI).

Nearly all study participants reported at least one type of sleep disturbance. This is not surprising given that we specifically recruited participants based on using sleep components of the Calm app in the last 90 days so that we would enrich the sample base with those attempting to improve sleep problems. Rates of sleep disturbances within the current sample were substantially higher than nationwide rates in the general population (estimated 20–40%), thus meeting our goals for a sleep-concerned population to survey [29]. Given the sleep concerns of the population recruited, it is also not surprising that nearly half of participants reported having at least one mental health condition when they first downloaded the Calm app. In particular, rates of anxiety (30%) and depression (27%) in the sample were substantially higher than general population rates (approximately 19% and 7%, respectively) [30]. It is well known that mental health problems are co-morbid conditions with sleep disturbances, and that there are bidirectional relationships between sleep quality and mental health [31].

Most participants reported that using Calm helped them to fall asleep, stay asleep, and get restful sleep. Additionally, higher frequency of using Calm was associated with perceived greater improvements in sleep. This finding provides preliminary support for the potential efficacy of Calm in addressing the unmet need of an effective, efficient, safe and convenient

**Table 10. Parameter estimates for regressions models of improvements in mental health diagnoses based on frequency of using Calm and its components, PSQI scores, and the interaction between frequency of use and PSQI (i.e., Step 2).**

| | | Anxiety | | | Depression | | | PTSD | | |
|---|---|---|---|---|---|---|---|---|---|---|
| | Predictor | N | Est. (SE) | p | N | Est. (SE) | p | N | Est. (SE) | p |
| Calm, overall | | 2,865 | | | 2,584 | | | 771 | | |
| | Intercept [threshold = 0] | | -2.18 (0.24) | < .001 | | -1.06 (0.25) | < .001 | | -0.65 (0.48) | .172 |
| | Intercept [threshold = 1] | | 0.82 (0.24) | .001 | | 1.53 (0.25) | < .001 | | 1.91 (0.48) | < .001 |
| | Usage frequency | | 0.21 (0.04) | < .001 | | 0.26 (0.04) | < .001 | | 0.35 (0.09) | < .001 |
| | PSQI | | -0.11 (0.03) | < .001 | | -0.07 (0.03) | .007 | | -0.06 (0.05) | .186 |
| | Usage x PSQI | | -0.00 (0.01) | .969 | | -0.01 (0.01) | .152 | | -0.01 (0.01) | .146 |
| Sleep Stories | | 2,615 | | | 2,383 | | | 706 | | |
| | Intercept [threshold = 0] | | -3.44 (0.18) | < .001 | | -2.29 (0.18) | < .001 | | -2.24 (0.35) | < .001 |
| | Intercept [threshold = 1] | | -0.54 (0.16) | .001 | | 0.19 (0.17) | .266 | | 0.30 (0.34) | .387 |
| | Usage frequency | | -0.06 (0.04) | .104 | | 0.02 (0.04) | .688 | | 0.06 (0.08) | .426 |
| | PSQI | | -0.14 (0.02) | < .001 | | -0.11 (0.02) | < .001 | | -0.13 (0.04) | < .001 |
| | Usage x PSQI | | 0.01 (0.00) | .054 | | -0.00 (0.00) | .871 | | -0.00 (0.01) | .943 |
| Sleep meditations | | 1,997 | | | 1,782 | | | 565 | | |
| | Intercept [threshold = 0] | | -3.30 (0.17) | < .001 | | -2.42 (0.17) | < .001 | | -2.72 (0.32) | < .001 |
| | Intercept [threshold = 1] | | -0.26 (0.14) | .075 | | 0.22 (0.15) | .147 | | -0.12 (0.29) | .671 |
| | Usage frequency | | 0.13 (0.05) | .007 | | 0.16 (0.05) | .001 | | 0.07 (0.08) | .420 |
| | PSQI | | -0.12 (0.02) | < .001 | | -0.12 (0.02) | < .001 | | -0.16 (0.03) | < .001 |
| | Usage x PSQI | | 0.00 (0.01) | .940 | | -0.00 (0.01) | .594 | | 0.00 (0.01) | .798 |
| General meditations | | 1,872 | | | 1,660 | | | 524 | | |
| | Intercept [threshold = 0] | | -3.01 (0.19) | < .001 | | -2.26 (0.18) | < .001 | | -2.58 (0.36) | < .001 |
| | Intercept [threshold = 1] | | 0.07 (0.16) | .653 | | 0.51 (0.17) | .003 | | 0.03 (0.33) | .919 |
| | Usage frequency | | 0.19 (0.04) | < .001 | | 0.21 (0.05) | < .001 | | 0.08 (0.08) | .319 |
| | PSQI | | -0.10 (0.02) | < .001 | | -0.11 (0.02) | < .001 | | -0.15 (0.03) | < .001 |
| | Usage x PSQI | | -0.00 (0.01) | .547 | | -0.00 (0.01) | .658 | | -0.00 (0.01) | .664 |
| Music/soundscapes | | 1,951 | | | 1,793 | | | 555 | | |
| | Intercept [threshold = 0] | | -2.95 (0.17) | < .001 | | -2.13 (0.17) | < .001 | | -2.11 (0.32) | < .001 |
| | Intercept [threshold = 1] | | -0.10 (0.15) | .517 | | 0.41 (0.16) | .009 | | 0.48 (0.31) | .119 |
| | Usage frequency | | 0.10 (0.04) | .022 | | 0.12 (0.04) | .005 | | 0.19 (0.08) | .023 |
| | PSQI | | -0.10 (0.02) | < .001 | | -0.09 (0.02) | < .001 | | -0.10 (0.03) | .001 |
| | Usage x PSQI | | -0.00 (0.00) | .743 | | -0.01 (0.00) | .218 | | -0.01 (0.01) | .214 |

**Table 11. Parameter estimates for regressions models of improvements in mental health diagnoses based on PSQI scores and frequency of using all Calm components.**

| | Model | | | | | |
|---|---|---|---|---|---|---|
| | Anxiety (N = 1,403) | | Depression (N = 1,280) | | PTSD (N = 413) | |
| Predictor | Est. (SE) | p | Est. (SE) | p | Est. (SE) | p |
| Intercept [threshold = 0] | -2.88 (0.19) | < .001 | -2.11 (0.18) | < .001 | -2.30 (0.35) | < .001 |
| Intercept [threshold = 1] | 0.15 (0.16) | .364 | 0.61 (0.17) | < .001 | 0.43 (0.33) | .186 |
| Sleep Stories usage | 0.02 (0.02) | .493 | 0.04 (0.02) | .113 | 0.06 (0.04) | .124 |
| Sleep meditation usage | 0.07 (0.03) | .018 | 0.04 (0.03) | .207 | 0.03 (0.05) | .520 |
| General meditation usage | 0.16 (0.02) | < .001 | 0.19 (0.03) | < .001 | 0.10 (0.04) | .021 |
| Music/soundscape usage | 0.04 (0.02) | .109 | 0.01 (0.02) | .579 | 0.03 (0.04) | .453 |
| PSQI | -0.11 (0.01) | < .001 | -0.13 (0.02) | < .001 | -0.16 (0.03) | < .001 |

sleep tool for the millions of adults who suffer from sleep disturbance. Future randomized controlled trials are necessary to formally test the efficacy of Calm against an attention placebo control, in improving sleep disturbances.

Due to the diversity of sleep content available on meditation mobile apps, users may choose sleep content based on their preferences (i.e., fall asleep to a meditation vs. listening to soundscapes). Sleep Stories were the most popular component of the Calm app for sleep, but all sleep components were associated with improvements in most types of sleep disturbances. Sleep Stories are based on mindfulness practices, but the format is substantially different from meditations, particularly with regard to instructions about posture, pace, and tone. Sleep Stories are fictional tales, somewhat similar to bedtime stories, that use mindfulness-inspired techniques, such as focusing on the breath and calling attention to sensation, to help listeners fall and stay asleep. There are clear theoretical models for how mindfulness meditation impacts health and sleep [16, 32], but there is no research regarding the mechanisms through which Sleep Stories may benefit users. Given the positive associations between using Sleep Stories and improvements in sleep disturbances, further research is needed to understand the mechanisms of action. However, it is also important to note that even when analyzed together (i.e., when controlling for use of other Calm components), all individual components of Calm were associated with improvements in sleep. Future studies are needed to understand the particular sleep components that elicit reductions in sleep disturbance.

There were two exceptions to the positive associations between using Calm sleep components and reporting reductions in sleep disturbance. First, using general meditations at night for sleep was not associated with improvements in falling asleep.

Second, when controlling for the use of other sleep components, those who used general meditations more often were significantly *less* likely to report that Calm helped them to fall asleep. This may be because meditation can increase attention and arousal when not in a sleep-specific context [33]. Unlike Sleep Stories and sleep meditations, general meditations guide the listener to sit upright and remain alert, focused, and attentive. While meditation in general has been shown to improve sleep and decrease pre-sleep arousal, most research has considered the broader effects of participating in meditation programs, not engaging in meditation immediately before sleep [34]. Although previous research has shown that many subscribers use Calm shortly before bed [23], future research would benefit from collecting additional information about the time of day that users engage with different types of content and how that may relate to its effectiveness for improving sleep.

Those with more severe sleep disturbance may benefit from using Calm for sleep. Research on the extent to which severity of insomnia or sleep disturbance impacts the effects of treatments for sleep is mixed [35]. However, there is a common belief among healthcare providers that more severe symptoms warrant higher levels of care, requiring the investment of more time and resources [36]. Findings from the current study suggest that even those with more severe sleep disturbance can benefit from "low-intensity" interventions that place fewer demands on resources (e.g., specialist time) [37], suggesting that apps such as Calm may be useful as adjunctive components to more intensive treatments, an early component in a stepped-care framework, or an as-needed aid for fluctuating symptoms or intermittent periods of sleep difficulty [38].

Among users who reported having mental health diagnoses (i.e., anxiety disorder, depression, PTSD), most reported that using Calm, in general, had helped to improve their condition or better manage their symptoms. Similar to our sleep findings, higher frequency of calm use was associated with more self-reported mental health improvements. However, we found differences among the sleep components that were related to these improvements. For example, using sleep meditations was associated with improvements in anxiety while using general

meditations was associated with improvements in both anxiety and depression. Using Sleep Stories and music/soundscapes was not associated with any mental health improvements. Importantly, using Calm in general, regardless of components, was associated with self-reported improvements in both sleep and mental health. These findings suggest the sleep components such as Sleep Stories, and music/soundscape may only help improve sleep, while meditations (general or sleep-specific) might be needed to improve mental health. More research testing specific content and how it effects sleep and mental health is warranted.

## Strengths and limitations

The current study is novel and addresses an important gap in the literature by providing information on the use of meditation mobile apps in a naturalistic setting. Research using commercially available apps may be more likely to reflect real-world behavior and, in the future, findings may more readily inform the development of practical, scalable interventions for sleep disturbance. This study also has several limitations. First, this was a cross-sectional survey. Answers were self-reported and retrospective. Given that recent literature has shown that there are often discrepancies between objectively logged and self-reported usage of digital media tools [39]. further corroboration of these findings with objective app-usage data would bolster confidence in their interpretation. Additionally, because the survey was developed specifically for this study, there is no external evidence to corroborate its reliability or validity. Second, while the study benefits from a large sample size, this was likely a highly engaged sample of Calm users as they had to have opened an email and used a sleep component of the app in the last 90 days. Questions regarding the helpfulness of Calm for sleep and mental health were unidirectional, such that participants did not have the opportunity to report deterioration of sleep or mental health. Third, our sample was mostly white, non-Hispanic females who were educated and mostly employed, limiting the generalizability of our findings. Finally, it should be noted that analyses were conducted with individuals who reported experiencing sleep disturbance at the time they downloaded Calm, whereas the PSQI assesses sleep disturbances during the past month and questions about the frequency of using Calm's sleep components were not limited to a specific time window. Therefore, results from the PSQI cannot be interpreted as reflective of sleep disturbance at the onset of Calm usage, and the moderation effects described must be considered as general relationships that are not necessarily sequential or time-specific. Future research would benefit from longitudinal analyses that can explicitly model these associations over time.

## Conclusions

Using a cross-sectional survey of subscribers of the mobile app Calm who endorsed sleep disturbances or used one of Calm's sleep components at least once in the past 90 days, we observed high rates of both sleep disturbance and mental health concerns. Higher frequency of using Calm was associated with higher perceived improvements in sleep disturbance and in mental health concerns. Sleep Stories were most commonly used for sleep, but all sleep-related app components were associated with perceived improvement in sleep disturbance. Only meditations were associated with self-reported improvement in mental health. Findings strongly support a future randomized controlled trial to test the efficacy of Calm, a potentially accessible and user-friendly solution to the current sleep epidemic.

## Author Contributions

**Conceptualization:** Jennifer Huberty, Linda Larkey, Ana-Maria Vranceanu, Michael R. Irwin.

**Data curation:** Megan E. Puzia.

**Formal analysis:** Megan E. Puzia.

**Investigation:** Jennifer Huberty.

**Methodology:** Jennifer Huberty, Linda Larkey, Ana-Maria Vranceanu, Michael R. Irwin.

**Project administration:** Megan E. Puzia.

**Supervision:** Jennifer Huberty.

**Writing – original draft:** Jennifer Huberty, Megan E. Puzia.

**Writing – review & editing:** Jennifer Huberty, Megan E. Puzia, Linda Larkey, Ana-Maria Vranceanu, Michael R. Irwin.

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
