## [Decision Letter · Decision Letter 0]

8 Jul 2021

PONE-D-21-08707

Can a meditation app help my sleep? A cross-sectional survey of Calm users.

PLOS ONE

Dear Dr. Puzia,

Thank you for submitting your manuscript to PLOS ONE. After careful consideration, we feel that it has merit but does not fully meet PLOS ONE’s publication criteria as it currently stands. Therefore, we invite you to submit a revised version of the manuscript that addresses the points raised during the review process.

First of all, apologies for the excessive time spent in the first round of reviews of your paper. It has been really difficult to allocate suitable reviewers for this submission, as the topic covered is very specific.

With the aim of expediting the process, I am submitting the review report of the referee that already sent their comments and suggestions. Although there are some minor revisions to perform, two of the comments address critical issues of the paper. Therefore, please try to cover and respond to all them as good as possible.

We look forward to receiving your revised manuscript.

Kind regards,

Sergio A. Useche, Ph.D.

Academic Editor

PLOS ONE

Journal Requirements:

JH is currently the Director of Science at Calm. JH has been conducting research with Calm as a partner almost 5 years before becoming the Director of Science and the Scientific Advisory Board. AMV, LL, and MI are members of Calm’s Scientific Advisory Board and are independent from Calm leadership. Their role is to ensure the quality of Calm’s science. There are no financial incentives from the growth of Calm to any author.

Reviewers' comments:

Reviewer's Responses to Questions

**Comments to the Author**

1. Is the manuscript technically sound, and do the data support the conclusions?

Reviewer #1: Yes

2. Has the statistical analysis been performed appropriately and rigorously? 

Reviewer #1: Yes

3. Have the authors made all data underlying the findings in their manuscript fully available?

Reviewer #1: Yes

4. Is the manuscript presented in an intelligible fashion and written in standard English?

Reviewer #1: Yes

5. Review Comments to the Author

Reviewer #1: The authors report on a large cross-sectional survey (N = 9907) of Calm users, examining associations between app usage and improvements in different aspects of sleep quality. Sleep components of the app were associated with reductions in sleep disturbance, whereas general meditation was not. Greater app usage was associated with more improvement. Greater severity of sleep disturbance was associated with more benefit from using the app. Participants also reported improvements in mental health (anxiety, depression).

I found this to be a comprehensive secondary analysis of an interesting dataset, and it should be quite informative to those in the fields of digital wellness and sleep. The paper was clearly written, and an enjoyable read. I have just a few questions and suggestions.

1) I’m a bit confused about the administration of the PSQI. Participants were eligible for the study if they completed at least one session of Calm in the last 90 days, but the PSQI assesses sleep disturbances over the previous month, which means that data may reflect sleep quality before, during, or after the period of app usage. If I’m correct about that, a more nuanced discussion of the moderating effects of sleep quality is warranted, since does not necessarily reflect that improvement is associated poorer sleep at baseline.

2) Why did the authors choose to analyze self-reported app usage instead of directly accessing usage logs? The recent Parry et al. (2021; Nature Human Behaviour) paper highlights that associations between self-reported and log-based measures may be low, and the paper might be enriched if the researchers studied objective data instead.

3) Lines 307-309: “To our knowledge, this is the first study to explore perceived improvements in sleep and mental health…”. I suggest removing this: the authors have published previous reports on Calm usage and there are reports that other apps (e.g. Headspace; Flett et al., 2020, Psychology and Health) have effects in reducing distress and other symptoms.

4) Related to point #2, were there any available data on the time-of-day that general meditations were used? If these were often used at night, that would lend support to the argument made in the paragraph starting line 349 (about general meditation increasing arousal and attention)

6. PLOS authors have the option to publish the peer review history of their article (what does this mean?). If published, this will include your full peer review and any attached files.

Reviewer #1: No

---

## [Author Response · Author response to Decision Letter 0]

16 Aug 2021

PONE-D-21-08707

Can a meditation app help my sleep? A cross-sectional survey of Calm users.

Response to reviewers

Dear Dr. Useche,

Thank for you the opportunity to submit our revised manuscript to be considered for publication at PLOS ONE. We appreciate the time that you and the reviewer have dedicated to providing feedback on our submission. We have responded to reviewer comments below and revised the manuscript in line with their suggestions. Please see below for point-by-point responses (italicized). 

Reviewer comments

The authors report on a large cross-sectional survey (N = 9907) of Calm users, examining associations between app usage and improvements in different aspects of sleep quality. Sleep components of the app were associated with reductions in sleep disturbance, whereas general meditation was not. Greater app usage was associated with more improvement. Greater severity of sleep disturbance was associated with more benefit from using the app. Participants also reported improvements in mental health (anxiety, depression).

I found this to be a comprehensive secondary analysis of an interesting dataset, and it should be quite informative to those in the fields of digital wellness and sleep. The paper was clearly written, and an enjoyable read. I have just a few questions and suggestions.

Thank you for taking the time to read and provide feedback on this paper. We appreciate your thoughtful comments and have addressed your remaining questions below.

1) I’m a bit confused about the administration of the PSQI. Participants were eligible for the study if they completed at least one session of Calm in the last 90 days, but the PSQI assesses sleep disturbances over the previous month, which means that data may reflect sleep quality before, during, or after the period of app usage. If I’m correct about that, a more nuanced discussion of the moderating effects of sleep quality is warranted, since does not necessarily reflect that improvement is associated poorer sleep at baseline.

Thank you for bringing this up, and we agree that this is an important point. We have reviewed and revised the results and discussion section to limit suggestion that our findings provide direct support for any specific temporal relationships or imply prescription of meditation apps at specific points in the trajectories of sleep disturbance or sleep-disturbance treatment. We have also added a section to the limitations section to note this explicitly and cite the need for future longitudinal research in this area.

2) Why did the authors choose to analyze self-reported app usage instead of directly accessing usage logs? The recent Parry et al. (2021; Nature Human Behaviour) paper highlights that associations between self-reported and log-based measures may be low, and the paper might be enriched if the researchers studied objective data instead.

We agree that the paper is limited by our use of self-reported retrospective data, and thank you for sending the citation. We've added a sentence to the limitations to note this and cited the paper that you mentioned.

3) Lines 307-309: “To our knowledge, this is the first study to explore perceived improvements in sleep and mental health…”. I suggest removing this: the authors have published previous reports on Calm usage and there are reports that other apps (e.g. Headspace; Flett et al., 2020, Psychology and Health) have effects in reducing distress and other symptoms.

After reviewing/re-reviewing several of these papers, we agree and have removed this sentence from the paper.

4) Related to point #2, were there any available data on the time-of-day that general meditations were used? If these were often used at night, that would lend support to the argument made in the paragraph starting line 349 (about general meditation increasing arousal and attention)

We have additional self-report data on the time of day that Calm was most frequently used for sleep, published in the original paper with these survey data; however, most participants endorsed multiple responses and we do not know specifically which components were used at particular times. We have referenced the original paper in the discussion as it relates to this point, and noted that future research would benefit from collecting this information.

---

## [Decision Letter · Decision Letter 1]

6 Sep 2021

Can a meditation app help my sleep? A cross-sectional survey of Calm users.

PONE-D-21-08707R1

Dear Dr. Puzia,

We’re pleased to inform you that your manuscript has been judged scientifically suitable for publication and will be formally accepted for publication once it meets all outstanding technical requirements.

Kind regards,

Sergio A. Useche, Ph.D.

Academic Editor

PLOS ONE

Additional Editor Comments (optional):

Reviewers' comments:

Reviewer's Responses to Questions

**Comments to the Author**

1. If the authors have adequately addressed your comments raised in a previous round of review and you feel that this manuscript is now acceptable for publication, you may indicate that here to bypass the “Comments to the Author” section, enter your conflict of interest statement in the “Confidential to Editor” section, and submit your "Accept" recommendation.

Reviewer #1: All comments have been addressed

2. Is the manuscript technically sound, and do the data support the conclusions?

Reviewer #1: Yes

3. Has the statistical analysis been performed appropriately and rigorously? 

Reviewer #1: Yes

4. Have the authors made all data underlying the findings in their manuscript fully available?

Reviewer #1: Yes

5. Is the manuscript presented in an intelligible fashion and written in standard English?

Reviewer #1: Yes

6. Review Comments to the Author

Reviewer #1: Thanks you for your revision, which was responsive to my concerns. I look forward to seeing the paper in print.

7. PLOS authors have the option to publish the peer review history of their article (what does this mean?). If published, this will include your full peer review and any attached files.

Reviewer #1: No

---

## [Editor Report · Acceptance letter]

14 Oct 2021

PONE-D-21-08707R1 

Can a meditation app help my sleep? A cross-sectional survey of Calm users. 

Dear Dr. Puzia:

I'm pleased to inform you that your manuscript has been deemed suitable for publication in PLOS ONE. Congratulations! Your manuscript is now with our production department. 

Kind regards, 

on behalf of

Dr. Sergio A. Useche 

Academic Editor

PLOS ONE